# EphrinA1-Fc Attenuates Ventricular Remodeling and Dysfunction in Chronically Nonreperfused WT but not EphA2-R-M mice

**DOI:** 10.3390/ijms21165811

**Published:** 2020-08-13

**Authors:** K’Shylah S. Whitehurst, Victoria A. Chan, Heather K. Estes, Smrithi Valsaraj, Susan Kent, Uma M. Sharma, R. Christopher Chase, Maliha Bhuiyan, Jitka A. I. Virag

**Affiliations:** Department of Physiology, Brody School of Medicine, East Carolina University, Greenville, NC 27834, USA; whitehurstk14@students.ecu.edu (K.S.W.); chanv17@students.ecu.edu (V.A.C.); heather.k.estes@gmail.com (H.K.E.); valsarajs13@students.ecu.edu (S.V.); susandkent@gmail.com (S.K.); sunuma@gmail.com (U.M.S.); chaser17@students.ecu.edu (R.C.C.); maliha.bhuiyan@gmail.com (M.B.)

**Keywords:** myocardial infarction, remodeling, fibrosis, EphA2, ephrinA1

## Abstract

Background: EphrinA1-Fc abolishes acute I/R injury and attenuates nonreperfused cardiac injury 4 days after permanent occlusion in mice. The goal of this study was to assess the capacity of a single intramyocardial administration of ephrinA1-Fc at the time of coronary artery ligation, to determine the degree to which early salvage effects translate to reduced adverse remodeling after 4 weeks of nonreperfused myocardial infarction (MI) in wild-type B6 and EphA2-R-M (EphA2 receptor null) mice. Methods: At 4 weeks post-MI, echocardiography, histologic and immunohistochemical analyses of B6 mouse hearts were performed. Primary mouse cardiac fibroblasts (FBs) isolated from B6 mice cultured in the presence of low and high dose ephrinA1-Fc, both with and without pro-fibrotic TGF-β stimulation and Western blots, were probed for relative expression of remodeling proteins MMP-2, MMP-9 and TIMP-1, in addition to DDR2 and (p)SMAD2/3/totalSMAD2/3. Results: EphrinA1-Fc preserved a significant degree of contractile function, decreased adverse left ventricular remodeling, attenuated excessive compensatory hypertrophy, and decreased interstitial fibrosis in wild-type (WT) B6 mouse hearts. In contrast, most of these parameters were poorer in ephrinA1-Fc-treated EphA2-R-M mice. Of note, fibrosis was proportionately decreased, implying that other EphA receptor(s) are more important in regulating the pro-fibrotic response. Primary FBs showed disparate alteration of MMP-2, MMP-9 and TIMP-1, as well as DDR2 and p-SMAD2/3/totalSMAD2/3, which indicates that matrix remodeling and cardiac fibrosis in the injured heart are influenced by ephrinA1-Fc. Conclusion: This study demonstrates the capacity of a single administration of ephrinA1-Fc at the onset of injury to attenuate long-term nonreperfused post-MI ventricular remodeling that results in progressive heart failure, and the important role of EphA2 in mitigating the deleterious effects.

## 1. Introduction

Irreversible tissue damage resulting from an acute myocardial ischemic (AMI) event causes the inflammation and structural remodeling of the ventricle, leading to progressive dilation, dysfunction, and ultimately heart failure. The manipulation of early cell signaling events to mitigate injury and the subsequent onset of adverse remodeling represents a potential therapeutic approach to reducing the severity of the initial ischemic insult, and thereby averting the inevitable development of heart failure; a socioeconomic crisis for which we still have no clinically meaningful treatment.

The Eph receptors (Eph-R) and their ligands, the ephrins, are the largest family of receptor tyrosine kinases, consisting of A and B subclasses which together have 16 receptors (EphA1-10 and Eph B1-6) and nine ligands (ephrinA1-5 and ephrinB1-4). Ephrin-Eph-R signaling influences cytoskeletal dynamics to effect cell differentiation, proliferation and migration during development [1,2], and plays a pivotal role in various cancers [3,4,5]. EphrinA1, an angiogenic protein stimulated by cytokines, interacts with the EphA2-R to promote tumor development [1,6,7,8,9,10]. EphA2-R is also involved in inflammation, as demonstrated by the increased sensitivity of EphA2-R-deficient mice to allergens caused by increased endothelial cell priming and immune cell extravasation, leading to increased tissue inflammation [11,12,13,14].

Our lab and others have reported that ephrinA1-EphA-R signaling can be manipulated to preserve cardiomyocyte function after MI without alterations in vascularity. In doxazocin-treated HL-1 cells, increased EphA2-R expression and reduced phosphorylation in response to lithocholic acid protected cells from apoptosis [15]. We have shown that a single injection of ephrinA1-Fc is cardioprotective in murine models of both acute I/R and chronically, nonreperfused MI at 4 days post-MI [16,17,18]. In EphA2-R-M hearts, we reported that chronically nonreperfused ischemia resulted significant cardiomyocyte damage and inflammation, leading to alterations in ventricular remodeling, interstitial fibrosis, and cardiac dysfunction [17,19,20]. We now propose that a single administration of ephrinA1-Fc at the time of permanent coronary artery occlusion will provide protection in WT mice that endures beyond the initial injury. We expect this will be evidenced by the attenuated dysfunction and amelioration of associated remodeling parameters; however, since EphA2-R is pivotal to several aspects of cellular function that determine the protective effects, EphA2-R-M mice lacking functional EphA2-R will derive little to no benefit from the same treatment. Additionally, the analysis of changes in MMP-2 and -9, TIMP-1, DDR2 and p-SMAD2/3/total SMAD2/3 in response to the treatment of isolated primary cardiac fibroblasts from WT mice with ephrinA1-Fc with or without TGF-β will shed light on the pathways involved in ephrinA1-Fc/EphA2-R in extracellular matrix remodeling. These studies will provide the foundation for further studies to optimize the use of ephrinA1-Fc in alleviating injury even in the absence of reperfusion, thus forestalling the deleterious progression to heart failure.

## 2. Results

### 2.1. Survival Data

Animals were monitored daily for the first week after surgery experiment, and then twice per week thereafter. Mice were all observed to be bright, alert and responsive, and none were euthanized prematurely. Deaths only occurred during the first week post-MI and were due to spontaneous incidence, likely arrhythmia. The differences in survival among the four MI groups (sham controls of both strains experienced 100% survival; data not shown) are presented in a Kaplan–Meier graph in Figure 1 (only the first week is shown for clarity since those mice survived to week 4). The survival of WT B6 mice with ephrinA1-Fc treatment was 92%, compared to 57% for EphA2-R-M post-MI with ephrinA1-Fc treatment. In WT B6 infarcted mice treated with IgG-Fc, survival was 79%, compared to the EphA2 infarcted mice treated with IgG-Fc, of which 71% survived.

### 2.2. Morphometry

Representative images of myocardial cross sections acquired at 20× are shown in Figure 2A–F. left ventricular (LV) parameters, LV chamber diameter (LVID_d_ and LVID_s_), LV average wall thickness (AWT), LV septal wall thickness (SWT) and LV Volume (LVV_d_ and LVV_s_) in systole (_s_) and diastole (_d_) were determined, and the averages are summarized in Table 1. The LVID_d_ and LVID_s_ were increased in both WT B6 and EphA2-R-M at 4 weeks post-MI compared to uninjured control (* *p* < 0.05), but there was no difference in these dimensions between any of the 4 week groups. The LV anterior wall thickness (AWT) exhibited a nonsignificant trend (*p* = 0.0836) towards increased thickness in WT 4 week, as compared with EA1-treated, but there was no difference between these groups in the EphA2-R-M mice. Interestingly, septal wall thickness (SWT) was significantly increased in the EA1-treated EphA2-R-M group as compared to the untreated EphA2-R-M 4 week MI. In the WT B6 4 week untreated mice, LVV_d_ and LVV_s_ decreased in the 4 week ephrinA1-Fc treated mice (^†^
*p* < 0.01), but both were unaltered by the EA1 treatment in the EphA2-R-M mouse hearts. Lastly, the left ventricular chamber area was not different between the 4 week untreated and 4 week EA1-treated WT mice, but it was larger in the ephrinA1-Fc-treated EphA2-R-M as compared with the EphA2-R untreated 4 week control group (^‡^
*p* = 0.0017).

Given the variability in these parameters, we calculated the Expansion Index (LV_chamber_/LV_area_ × septal wall thickness/free wall thickness; Figure 2G) and found that it was not different between WT and EphA2 controls (0.17 ± 0.01 and 0.21 ± 0.2, respectively), but in WT mice, it increased nearly 3-fold post-MI (0.61 ± 0.08) and was returned to near normal values in ephrinA1-Fc-treated mice (0.4 ± 0.1). In contrast, in the EphA2-R-M 4 week mice, the expansion index tended to be worse in ephrinA1-Fc-treated mice, but was not significantly different (0.26 ± 0.02 and 0.58 ± 0.21).

### 2.3. Myocyte Cross-Sectional Area

Representative images of myocyte cross-sectional areas (MCSA, Figure 3A–F) of the WT epicardial and endocardial were not different within each group, and are thus presented as average of the two populations (Figure 3G). The MCSA in control WT B6 mice was 208.5 ± 8.3. After 4 weeks post-MI, it increased slightly but not significantly to 239.7 ± 9.4, and was actually lower (150 ± 5.6 in ephrinA1-Fc-treated mouse hearts). The MCSA in EphA2 uninjured control mice was 233.4 ± 8.9. Similar to the slight increase in WT mice after 4 weeks post-MI, the MCSA in EphA2 4 week mice increased to 268 ± 14.5. In contrast, at 4 weeks post-MI in EphA2-R-M 4 week ephrinA1-Fc-treated mice, the MCSA increased nearly two-fold to 520.5 ± 45. (*p* < 0.0001).

### 2.4. Fibrosis

Representative images of interstitial fibrosis (Figure 4A–F), calculated as a percentage of the LV (red/(green + red − white)), shows that uninjured WT (*n* = 10) and EphA2-R-M (*n* = 10) hearts (0.74 ± 0.06% and 0.63 ± 0.06%, respectively) were not significantly different (Figure 4G). At 4 weeks post-MI, WT (*n* = 9) and EphA2-R-M (*n* = 10) fibrosis increased by 56% and 51% from control levels, respectively, with 1.66 ± 0.11% and 1.29 ± 0.15% of the tissues containing collagen fibers in the interstitium remote to the infarct in the same transverse plane (*p* < 0.001). In response to the ephrinA1-Fc treatment administered at the time of left anterior descending (LAD) artery occlusion, the WT (*n* = 11) heart was 35% less fibrotic (1.075 ± 0.12; *p* = 0.0049) whereas the EphA2-R-M (*n* = 4) heart was 33% less fibrotic, but this was not significantly different (0.86 ± 0.29%) from the untreated EphA2-R-M 4 week untreated heart.

### 2.5. Cardiac Function

No differences in heart rate were observed in conscious WT B6 or EphA2-R-M mice in any of the treatment groups (range: 593 ± 22−633 ± 29 bpm). The ejection fraction and representative traces are shown in Figure 5. The baseline ejection fraction in WT (*n* = 11) mice was 95 ± 1%, compared with 79 ± 2% in EphA2-R-M (*n* = 22) mice (*p* < 0.001). The ejection fractions at 4 weeks post-MI in WT (*n* = 5) and EphA2-R-M (*n* = 12) mice were 65 ± 4% and 53 ± 4%, respectively (*p* = 0.04). Notably, in the 4 week ephrinA1-Fc treated mice (*n* = 11), ejection fraction (EF) was 76 ± 4%, or 11% higher compared to both the untreated group and the ephrinA1-Fc-treated EphA2-R-M group (*n* = 4; 65 ± 4%), which was not significantly different from the 4 week untreated EphA2-R-M group (54 ± 7%).

Fractional shortening (FS) followed similar trends (data not shown). In uninjured sham control mice, FS was 69 ± 2 and 47 ± 1 in EphA2-R-M mice. At 4 weeks, the FS values were 35 ± 3 and 27 ± 2 for untreated WT and EphA2-R-M, whereas for ephrinA1-Fc-treated mice they were 43 ± 2, and 28 ± 4 for WT and EphA2-R-M, respectively.

### 2.6. Immunohistochemical Detection of α-SMA^+^ Myofibroblasts in Infarcted Mouse Heart

Representative immunohistochemical images (Figure 6) of smooth muscle α-actin positive cells in uninjured B6 and EphA2 control heart tissue demonstrate expected staining of vascular smooth muscle cells (VSMCs) in large vessels. At 4 weeks post-MI, SMα-A positive myofibroblasts are visible in the infarcted myocardium in both B6 and EphA2, but to a much lesser extent in ephrinA1-Fc-treated B6 and EphA2-R-M mice (*n* = 3/group).

### 2.7. Western Blotting Analysis of Remodeling Indices in Isolated Primary Cardiac Fibroblasts

Figure 7 depicts the relative changes in expression levels of the pro and the active forms of MMP-9, MMP-2 and TIMP-1 enzymes in ephrinA1-Fc-treated +/TGF-β primary cardiac fibroblasts, showing that they are modulated in a dose-dependent manner (*n* = 3/group; *p* < 0.05: * vs. control ^†^ vs. 0.1 µM ephrinA1-Fc ^‡^ vs. TGF-β). The low dose of ephrinA1-Fc alone exhibited a non-significant tendency to increase DDR2 expression, and was not further enhanced by the high dose of TGF-β alone, but the combination of either low or high dose ephrinA1-Fc with TGF-β appeared to bring DDR2 expression back down to the level of the untreated control. While the low dose of ephrinA1-Fc tended to increase p-SMAD2/3/SMAD2/3, the higher dose attenuated their expression in primary cardiac fibroblasts, whereas the high dose reduced the expression of these compared to TGF-β alone, as well as in combination with either the low or high dose of ephrinA1-Fc (*n* = 3/group).

## 3. Discussion

This is the first report to examine the long-term effects of a single intramyocardial injection of ephrinA1-Fc delivered at the onset of the ischemic event in the maintenance of cardiac structure and function following a permanent coronary occlusion 4 weeks post-MI. Specifically, we have demonstrated that ephrinA1-Fc attenuates the resultant ventricular remodeling, the pathologic compensatory hypertrophy, the interstitial fibrosis and the cardiac dysfunction normally observed at 4 weeks post-MI in WT mice. Moreover, this is not only lost in mice lacking the EphA2-R, but exacerbated, causing increased mortality. This suggests that ephrinA1-Fc/EphA2 signaling is an important determinant of survival and ventricular remodeling. Confirmatory evidence in isolated cardiac fibroblasts demonstrated that ephrinA1-Fc influences the expression of major matrix remodeling proteins MMP-2, MMP-9 and TIMP-1. Additionally, we have observed that ephrinA1-Fc reduces TGF-β-induced DDR2 expression and dose-dependently alters pSMAD2/3/totalSMAD 2/3, but not in the presence of TGF-β, providing evidence that ephrinA1-Fc does influence fibroblast activation but not via SMAD2/3 signaling under these conditions [21]. Further studies are needed to explore the role and timing of EphA2-R, and/or other EphA1-R activity, that mediates these effects leading to improved outcomes.

We have previously shown that the manipulation of ephrinA1-EphA-R signaling can alter the fate of the myocardium after acute infarction without changes in vascularity [17], and that the EphA2-R influences the severity of cardiomyocyte injury and early ventricular remodeling [19]. Our current findings suggest that ephrinA1-Fc treatment at the time of coronary artery ligation in a chronic model of nonreperfused MI attenuates cardiac dysfunction, excessive pathological cardiomyocyte hypertrophy, and deleterious fibrosis in WT B6 mice, but does not prevent these adverse changes in the infarcted EphA2-R-M myocardium. Specifically, left ventricular remodeling indices—LV area, free and septal wall thicknesses, expansion index and myocyte cross-sectional area—and interstitial fibrosis were all significantly increased in WT and EphA2-R-M hearts, whereas these deleterious remodeling effects were lessened by ephrinA1-Fc in WT mice. These data suggest that the reduction of early tissue injury by ephrinA1-Fc in WT mice reduced the reduction in EF by 11% (ephrinA1-Fc vs. IgG-Fc), providing substantial long-term protection and suggesting that the remaining 19% may be due to reperfusion injury (ephrinA1-Fc vs. control). In relation to the convention that MI size is 100% of the area at risk (AAR) in the absence of reperfusion, and is reduced by 50% with timely reperfusion, leaving MI size 25% of AAR with reperfusion, if we extrapolate proportionally, ephrinA1-Fc may reduce infarct injury up to 58% even in the absence of reperfusion; a significantly greater effect than that of reperfusion alone [22]. More comprehensive and precise means of measuring infarct size are required to confirm this.

The precise involvement of EphA2-R in regulating ischemic injury in the heart currently is unknown. This is due to the poorly understood signaling pathways influenced by EphA2-R, as well as the relative amount of expression in each of the multiple cell types. The few studies that have investigated EphA2-R have characterized its role in ischemic injury. For example, Thundyil et al. observed less inflammatory cell infiltrate and reduced apoptosis in EphA2-R-M mice with ischemic brain injury [23]. In contrast, the activation of EphA2-R protected HL-1 cells from apoptosis and renal ischemic injury in vivo [15,24]. These findings suggest that the role of EphA2-R in acute ischemic injury is a complicated tissue- and/or cell-specific phenomenon.

Acute myocardial injury is followed by inflammation, and the resolution phase is characterized by fibrosis, a molecularly and cellularly complex process that is a deleterious yet necessary component of cardiac remodeling [25]. Fibroblasts, the major cell type involved, are activated and stimulated post-MI to become myofibroblasts which migrate into the wound and produce collagen and other factors that ultimately replace necrotic tissue with a non-contractile, collagenous scar [26]. The weakening and stretching of this scar tissue imposed by the ventricular pressure and output demand ultimately leads to heart failure. The interaction between cardiomyocytes, inflammatory cells, myofibroblasts and vascular cells, as coordinated by remodeling enzymes such as MMPs and TIMPS, determines the nature of the extracellular matrix (ECM) and tensile strength of the scar [27,28,29]. Interestingly, membrane-anchored ephrinA1, expressed primarily by cardiomyocytes, can be proteolytically cleaved by MMP-2 and MMP-9 to release three functional monomeric products, but the role of these in the microenvironment of the infarcted heart is as yet unknown [30].

Our previous observation that there is more residual necrosis and decreased MMP-2 synthesis in EphA2-R-M hearts may be due to defective inflammatory cell function resulting from deficient NF-κB expression, leading to poorer remodeling and dysfunction. Although interstitial collagen deposition was proportionately reduced in EphA2-R-M mice, the observed pattern of distribution qualitatively appeared to be less organized, and so more investigation into the differential expression of collagen isoforms and the nature of the fibrillar structure is necessary in order to understand its role in the pathologic changes during ventricular wall thinning and infarct expansion. To investigate the role of ephrinA1-Fc/EphA-Rs on fibrosis, we isolated primary cardiac fibroblasts from WT hearts and exposed them to two doses of ephrinA1-Fc in the absence or presence of pro-fibrotic TGF-β, and measured pro- and active-MMP-2 and MMP-9 as well as TIMP-1. The results indicate that ephrinA1-Fc alone activates MMP-2, but this is abolished by TGF-β at the lower dose. Further, ephrinA1-Fc acts synergistically with TGF-β to increase MMP-9 activation. These data suggest that ephrinA1-Fc modulates the fibrotic response in post-MI myocardium, but the role of the other EphA-Rs differentially expressed on cells in cardiac tissue complicates the dissection of their respective contributions. Although the exogenously administered ephrinA1-Fc is gone after 24 h post-MI, the maintenance of endogenously expressed ephrinA1 on cardiomyocytes may influence the progression of injury and inflammation in such a way as to reduce the pro-fibrotic stimuli. Since TIMP-1 is known to be stimulated by pro-inflammatory cytokines, and these are reduced in ephrinA1-Fc-treated WT hearts, the observed decrease in expression of TIMP-1 facilitates the activation of MMP-9 and -2, advocating for more rapid healing/turnover and/or improved deposition of collagen. Given the complicated nature and expression pattern of MMPs and TIMPs, targeted experiments in isolated cells and transgenic mice are needed to clarify the role of these enzymes in mediating the ephrinA1-Fc-mediated attenuation of ventricular remodeling, and the EphA-Rs that drive the response.

To further investigate the mechanism by which ephrinA1-Fc reduces fibrosis and promotes infarct repair, we also investigated changes in DDR2 and pSMAD2/3/SMAD2/3 expression in isolated murine cardiac fibroblasts treated with ephrinA1-Fc with and without TGF-β. DDR2 is a fibroblast-specific, collagen-induced membrane-bound receptor tyrosine kinase that is upregulated in fibroblasts in response to TGF-β to influence the proliferation and motility of cardiac fibroblasts [31]. In rat vascular smooth muscle cells, the administration of recombinant TGF-β increased DDR2 expression in a manner similar to that induced by cyclic stretch [32], and in murine skin fibroblasts, deletion of DDR2 significantly reduced migration and MMP-2 activity, thus decreasing collagen deposition [33]. In the current study, the low dose of ephrinA1-Fc alone increased DDR2 expression as well as (p)SMAD2/3/SMAD2/3 expression in primary cardiac fibroblasts, whereas the high dose reduced the expression of these compared to TGF-β alone or with either the low or high dose of ephrinA1-Fc. In terms of TGF-β signaling, the phosphorylation of SMAD2/3 is pivotal in the transformation of FBs to the pro-fibrotic phenotype [34,35,36,37,38,39]. Taken together, these data indicate that ephrinA1-Fc promotes improved wound healing and reduces interstitial collagen deposition in vivo, at least in part, by modulating excessive and untimely pro-fibrotic signaling. However, the EphA-R(s) involved and the downstream intracellular signaling cascades are as yet unclear, and likely involve other pathways, such as MAPK/ERK signaling. This, in addition to reducing acute cardiomyocyte injury, establishes the ability of one injection of ephrinA1-Fc to regulate several aspects of wound repair; these multifunctional treatment properties are of critical importance in managing the timing and balance of factors that guide the complicated sequelae of infarct repair [16,17,20,40,41].

Lastly, the developmentally deficient vascularity and injury-induced angiogenic response of the EphA2-R-M mouse heart has been previously described in terms of both the uninjured and infarcted adult mouse heart. It is likely that the deficient and/or defective vessels are at least in part responsible for the compromised functional capacity of mice deficient of EphA2-R. This would result in a limited ability of the myocardium to respond to alterations in perfusion or oxygen concentration, which is necessary to promote normal wound healing. Although we have not performed functional analyses of the vessels or determined whether this deficiency alters perfusion, it is possible that inadequate revascularization may partially explain our findings in EphA2-R-M. In the current study, this is likely to be partly, but not exclusively, responsible for the poorer survival and remodeling characteristics that manifest in EphA2-R-M mice at 4 weeks post-MI. Moreover, since treated EphA2-R-M mice fare worse than the untreated group, there is probably compensation occurring via the activation/inhibition of other EphA-Rs that may be exaggerated. We have previously shown that, in contrast to the reduced expression of EphA6-R in WT mice, EphA6-R was upregulated in response to MI in EphA2-R-M. This may be due to the varying affinity of the ligands with the receptors, the resultant heterogeneous receptor binding and/or clustering, and subsequent activation/inhibition that culminate in downstream signaling cascades to modulate cell behavior [42]. Specifically, the activation of EphA2-R may effect cardioprotection, whereas EphA6-R binding in cardiomyocytes may be inhibitory, thus preventing potentially negative downstream effects. Additional investigation of the impact of EphA2-R and EphA6-R signaling in cardiomyocytes and other cardiac tissue cells is warranted to clarify their importance in cardiomyocyte survival, inflammation, revascularization and fibrosis following MI.

Our previous work revealing the cardioprotective benefits of endogenous ephrinA1-Fc administration in both acute I/R [16,18] and chronically nonperfused MI [17], as well as the severity of ischemic injury in both control and hyperglycemic EphA2-R-M mice [19,43], validate the critical importance of this bidirectional signaling system in mitigating several aspects of early injury. This sets the stage for the current study in which we demonstrate the robust effects of a single intramyocardial injection of ephrinA1-Fc in attenuation of remodeling in the chronically nonreperfused infarcted heart, and the pivotal role of EphA2-R-M in modulating this effect. The modulation of ephrinA1/EphA-R signaling represents an exciting new area of exploration in the heart, and further studies are needed to determine their role and therapeutic potential in order to maximally reduce both acute and chronic infarct injury, both in the presence and absence of reperfusion, as well as assess its potential applicability to other ischemic disease processes.

## 4. Materials and Methods

### 4.1. Animals

Experimental research protocols were approved 20 March 2018 by the East Carolina University Institutional Animal Care and Use Committee (IACUC) following the guidelines of the National Institutes of Health for the Care and Use of Laboratory Animals maintained by the PHS assurance (#A3469-01). B6129SF2/J mice (stock #101045) (WT) and B6129S6-Epha2^tm1Jrui^/J (stock #006028; a homozygous mutant; EphA2-R-M) mice (Jackson Labs, Bar Harbor, ME, USA) were used. All mice were housed in ventilated cages and animal care was maintained by the Department of Comparative Medicine at The Brody School of Medicine, East Carolina University. Mice were exposed to 12 h/12 h light/dark cycle conditions and received food and water ad libitum.

### 4.2. Surgical Procedure and Tissue Collection

Male WT and EphA2-R-M mice (8–12 weeks) were anesthetized with an intraperitoneal injection of 20 mL/g body weight Avertin (20 mg/mL) and mechanically ventilated. The left anterior descending coronary artery (LAD) was permanently occluded using an 8-0 suture. Subsequent blanching of the apical portion of the myocardium was evident and signified the successful occlusion and generating of a consistent infarct size and cTnI release as reported previously [17]. Immediately following ligation, 6 μL of 1.0 μg/uL ephrinA1-Fc (Sigma, E9902, St. Louis, MO, USA) or IgG-Fc (R&D systems, 110-HG-100; Minneapolis, MN, USA) as control was injected into the myocardium at the border zone using a 30 g sterile Hamilton syringe as described previously [16,17,18,44]. The tubes containing these compounds were color-coded by another lab member so that the operator was blinded and all subsequent analyses were performed prior to breaking the code. The rib cage, muscle and skin were then closed with a 6-0 suture. Four weeks after surgery, mice were anesthetized with a lethal intraperitoneal injection of 0.1 mL pentobarbital (390 mg/mL). At the time of tissue collection, a pneumothorax was performed and a cardiac stick using cold KCl (30 mM) was done to arrest the heart in diastole. The heart was subsequently resected at the aortic root, rinsed in PBS, and immersion-fixed in a zinc-based fixative or frozen in liquid nitrogen. Whole hearts cut transversely into 4 slices of equal thickness were processed (TP1020, Leica, Nußloch, Germany), and embedded in paraffin (Microm EC350, Richard-Allan Scientific, Kalamazoo, MI, USA). Sections of paraffin embedded tissues 5 µm thick were mounted on Superfrost Plus glass slides for histology or immunostaining.

### 4.3. Echocardiography and Blood Pressure

Echocardiography was performed on unanesthetized conscious mice [45]. A VisualSonics Vevo 2100 diagnostic ultrasound machine was used to obtain LV dimensions in diastole and systole using M-mode acquired at the mid-papillary level. Measurements (IVSd, LVPWd and LVIDd) were acquired in end-diastole at the point of maximal LV diastolic relaxation. Similarly, end-systolic dimensions (IVSs, LVPWs and LVIDs) were obtained during the most anterior systolic excursion of the LVPW that was associated with the smallest chamber dimension. Average measurements were derived from these traces using the leading-edge approach from 3 to 5 consecutive sinus beats, and standard calculation of ejection fraction (EF) was done using the formula: (LVIDd^3^–LVIDs^3^)/LVIDd^3^ × 100% [46].

Tail cuff blood pressure measurements using the Hatteras system (Cary, NC; 3 separate occasions within 5 days) showed no differences in average systolic pressure (126 ± 13–132 ± 16 mmHg), diastolic pressure (63 ± 9–84 ± 12 mmHg), mean arterial pressure (MAP) (81 ± 9–99 ± 11) or pulse (566 ± 27–621 ± 33 bpm).

### 4.4. Morphometry and Histology

Morphometric measurements were performed on heart tissue sections from the sham-operated control 4 weeks post-MI, and 4 weeks post-MI from the ephrinA1-Fc-treated WT and EphA2-R-M mice. Four H&E-stained (hematoxylin and eosin) section images of each heart were taken at 20× magnification using a DP70 digital camera. Two sections of heart tissue containing infarcted tissue located in the apical half, approximately 1 mm apart, were taken. Similarly, two sections of non-infarcted heart located in the basal half were taken. The left ventricular (LV) cross-sectional area was measured using Scion imaging software (v4.0.3.2, Scion Corporation, Frederick, MD, USA). LV anterior wall and septal thickness were measured using the average of 3–4 measurements in each of the 2 sections with infarct (excluding the border). Infarct size was measured using these four complete profiles of the transverse sections of each heart from apex to base. Infarct size/scar was computed as a percentage of the left ventricle.

Myocyte cross-sectional area (MCSA) was measured in 3–8 cardiomyocytes with centrally located nuclei in each of 6 images (400×) from both the endocardium and epicardium (*n* = 3–5/group) of viable areas from the two sections containing infarcted tissue using Scion imaging software.

Interstitial fibrosis was measured using a Picrosirus Red/Fast Green staining protocol. Briefly, samples were de-paraffinized and rehydrated, immersed in a 0.1% picrosirius red/fast green solution (Sigma-Aldrich, 365548, F7258, P6744; St. Louis, MO, USA) for 30 min, cleared and coverslipped [47]. Interstitial fibrosis was assessed in 4 images at 400× in 2 sections per heart using Adobe Photoshop to count the number of red and green pixels. These numbers were used to calculate percent collagen according to the following formula: number of red pixels/(red + green pixels − white) × 100%. The average for each animal was calculated and the data in the graph represent the average of 3–5 animals per group.

### 4.5. Immunostaining

Tissue sections were deparaffinized in xylene and endogenous peroxidases quenched with 3% H_2_O_2_ in methanol. Slides were rinsed in saline and incubated with antibodies for SMA (Dako, U7033; Via Real Carpintera, CA, USA) and DDR2 (Santa Cruz, sc-81707; Santa Cruz, CA, USA). Slides were incubated with the HRP-polymer kit, the reaction product was visualized with DAB (Vector Biolabs, SK-4100; Burlingame, CA, USA), counterstained with methyl green and dehydrated in xylene, and slides were coverslipped.

#### Isolation and Culture of Primary Mouse Cardiac Fibroblasts

Cardiac fibroblasts from five naïve B6 mice were isolated from whole LV minced under sterile conditions in a laminar flow cabinet. After the blood was rinsed, the suspensions were serially dissociated in 10% FBS media using a collagenase digestion buffer and trituration. After centrifugation, the supernatant was discarded, and cells were cultured in 5% CO_2_ at 37 °C and the media was changed every two days until the plates were 85% confluent. Cells from one animal did not grow enough to use and the results from the fourth were > 2xSD from the mean, and thus this outlier was removed from the analyses. Based on dosages of ephrinA1-Fc and TGF-β previously reported to treat cells in culture [15,34,48,49], the treatment groups were as follows: Control (vehicle), ephrinA1-Fc 0.1 µM or 1.0 µM, TGF-β 10 ng/mL, ephrinA1-Fc 0.1 µM + TGF-β 10 ng/mL, or ephrinA1-Fc 1.0 µM + TGF-β 10 ng/mL. These were applied to confluent primary mouse cardiac fibroblast cultures for 24 h. At the end of the experiment, protein from the cells was isolated for subsequent Western blotting.

### 4.6. Protein Extraction and Western Blotting

Protein was extracted from cells using Reagent 4 (Sigma, C0356; St. Louis, MO, USA). The Bradford Assay was used to quantify the amount of protein. Western blotting was performed on a 4–12% gradient Bis-Tris gel (BioRad; Hercules, CA, USA) in 1× MOPS running buffer. Thirty micrograms of protein extracted from cultured fibroblasts was loaded per well. The gel was run for 1 h at 155 V and transferred onto pure nitrocellulose membranes. Antibodies: MMP2 (PA1-16667), MMP9 (PA5-13199), TIMP1 (MA5-13688), DDR2 (NBP2-14927, Novus Biologicals, Centennial, CO, USA), (p)-SMAD2/3 (8828 and 8685, Cell Signaling Technologies; Danvers, MA, USA). Densitometry was performed using Image J software v.1.4.3.67 and the intensity of each protein was normalized to either GAPDH or total protein.

### 4.7. Statistics

The number of mice needed to determine statistical significance of each assay is indicated in the text and/or figure legends. Data were graphed using Excel and/or GraphPad Prism v8 software, and are presented as bar graphs or individual data points with the mean ± standard deviation. Statistical significance was defined as at least *p* < 0.05, and determined by Student’s *t*-tests or Two-way Analysis of Variance and Tukey’s post hoc analysis.

## Figures and Tables

**Figure 1 ijms-21-05811-f001:**
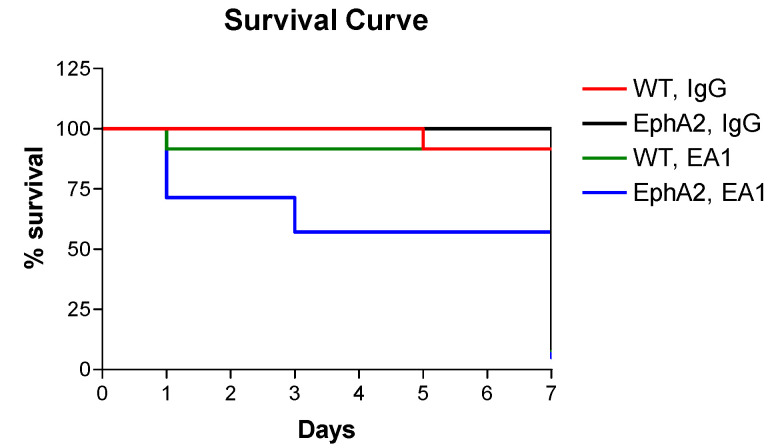
Kaplan–Meier Estimate Analysis Demonstrating Survival Rates in WT and EphA2-R-M Mouse Hearts. Kaplan–Meier graph showing survival rates of IgG-Fc (IgG; *n* = 8)- and ephrinA1-Fc (EA1; *n* = 11)-treated WT B6 and EphA2-R-M mice 1 week (same as 4 weeks) post-MI. These data show the poorer survival of EphA2-R mice treated with EA1 (*n* = 4) compared to IgG (*n* = 10) and both WT groups.

**Figure 2 ijms-21-05811-f002:**
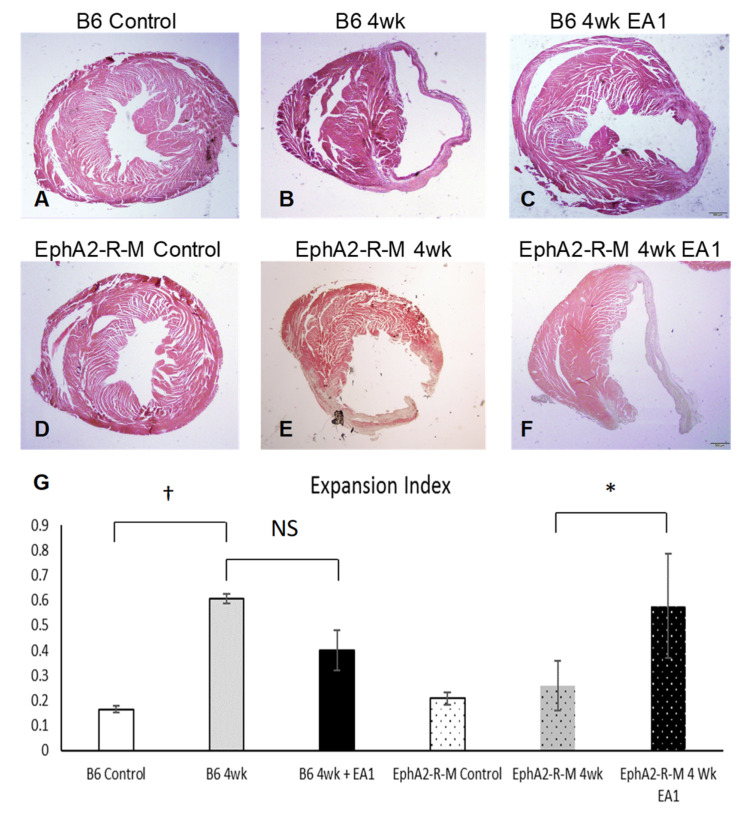
Hematoxylin and Eosin (H&E) Stains for Morphometric Analyses and Calculated Expansion Index in WT and EphA2-R-M Mouse Hearts. Representative H&E stains (20×) of WT (**A**–**C**) and EphA2-R-M (**D**–**F**) hearts used for morphometric determination of left ventricular remodeling indices (left ventricular area, scar area and thickness, chamber area) demonstrating that EA1 had a nonsignificant tendency to reduce adverse remodeling in WT B6 mouse hearts, but actually worsened infarct expansion (**G**) in EphA2-R-M mouse hearts. B6 WT control (*n* = 10), EphA2-R-M control (*n* = 10), WT B6 (IgG-Fc 4w, *n* = 8), EphA2-R-M (IgG-Fc, *n* = 10), EA1-treated WT B6 (*n* = 11) and ephrinA1-Fc-treated EphA2-R-M (*n* = 4). Statistical significance * *p* < 0.05 and ^†^
*p* < 0.01).

**Figure 3 ijms-21-05811-f003:**
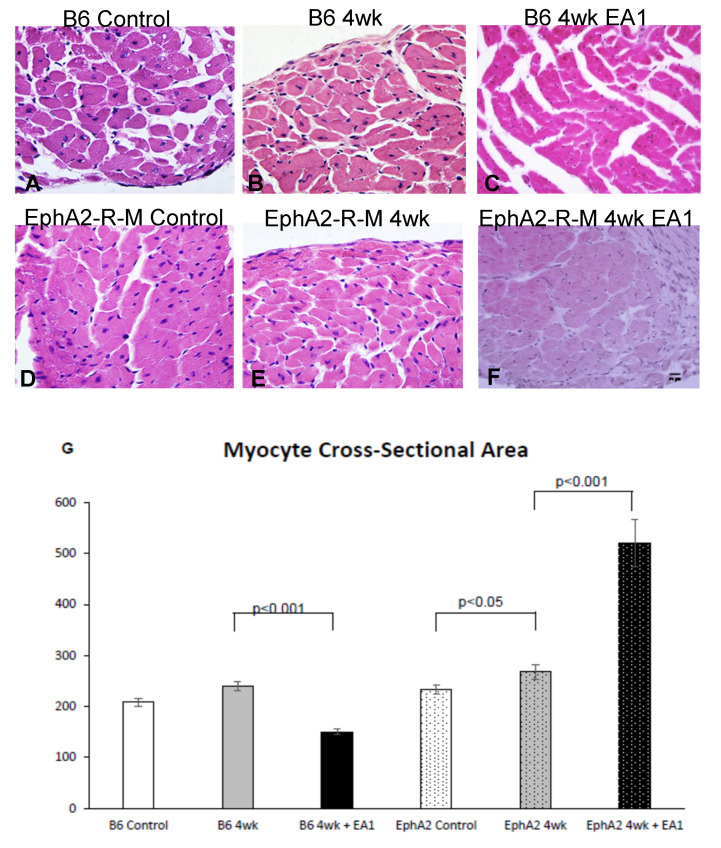
Myocyte Cross-Sectional Area in WT and EphA2-R-M Mouse Hearts. Myocyte cross-sectional area in WT and EphA2-R-M mice in uninjured control, untreated (IgG-Fc) vs. ephrinA1-Fc- treated mice at 4 weeks post-MI (stain: hematoxylin and eosin; 400×). Representative images in WT B6 (**A**–**C**) but not in EphA2-R-M (**D**–**F**) mouse hearts show that EA1 significantly attenuated compensatory hypertrophy (**G**), indicating that progression to the heart failure phenotype may be accelerated in mice deficient in EphA2-R. B6 WT control (*n* = 10), EphA2-R-M control (*n* = 10), WT B6 (IgG-Fc 4w, *n* = 8), EphA2-R-M (IgG-Fc, *n* = 10), EA1-treated WT B6 (*n* = 11) and ephrinA1-Fc-treated EphA2-R-M (*n* = 4).

**Figure 4 ijms-21-05811-f004:**
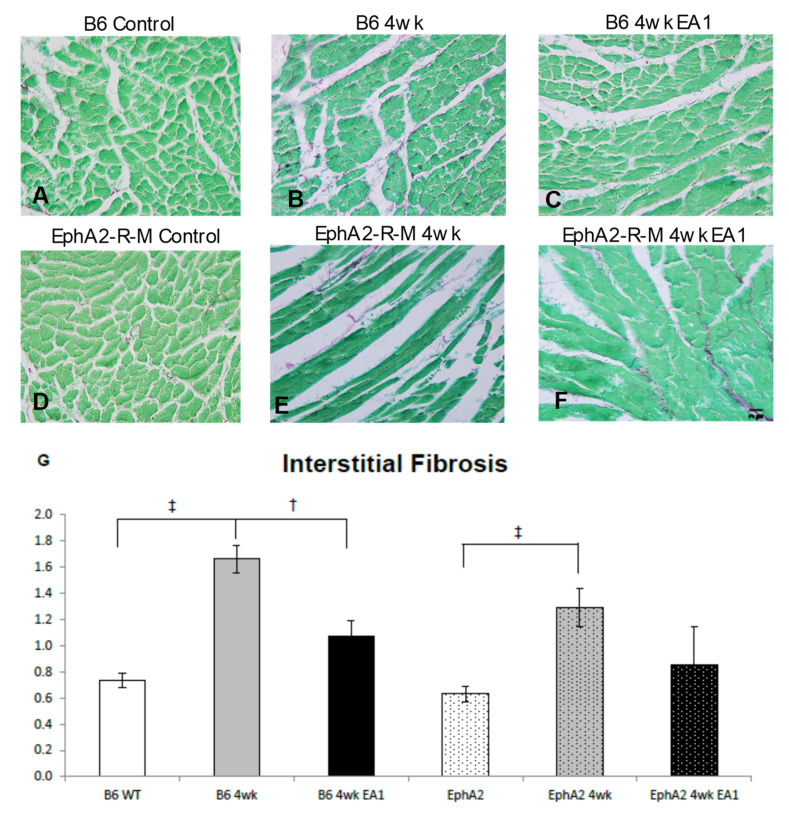
Interstitial Fibrosis in WT and EphA2-R-M Mouse Hearts. WT (**A**–**C**) and EphA2-R-M (**D**–**F**) images depicting interstitial fibrosis 4 weeks after infarction (stain: picrosirius red/fast green; 400x). The fibrotic response was attenuated by EA1 in WT B6 mouse hearts but not EphA2-R-M mouse hearts (**G**). B6 WT control (*n* = 10), EphA2-R-M control (*n* = 10), WT B6 (IgG-Fc 4w, *n* = 8), EphA2-R-M (IgG-Fc, *n* = 10), EA1-treated WT B6 (*n*= 11) and ephrinA1-Fc-treated EphA2-R-M (*n* = 4); ^†^
*p* < 0.05 and ^‡^
*p* < 0.001.

**Figure 5 ijms-21-05811-f005:**
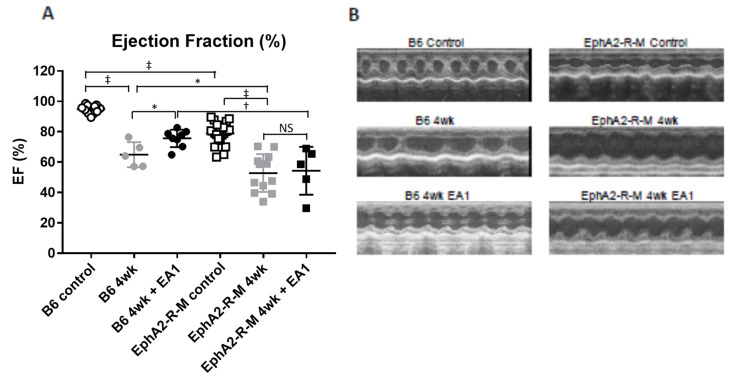
Cardiac Function in WT B6 and EphA2-R-M Mouse Hearts. EF (**A**) improved 10.7% from untreated to ephrinA1-Fc- treated B6 mice, but there was no statistically significant difference between treated and untreated EphA2 mice (* *p* < 0.05, ^†^
*p* < 0.01, ^‡^
*p* < 0.001). B6 WT control (*n* = 11), EphA2-R-M control (*n* = 22), WT B6 (IgG-Fc 4w, *n* = 12), EphA2-R-M (IgG-Fc, *n* = 13), EA1-treated WT B6 (*n* = 11) and ephrinA1-Fc-treated EphA2-R-M (*n* = 4). (**B**) Representative traces for all WT and EphA2-R-M groups.

**Figure 6 ijms-21-05811-f006:**
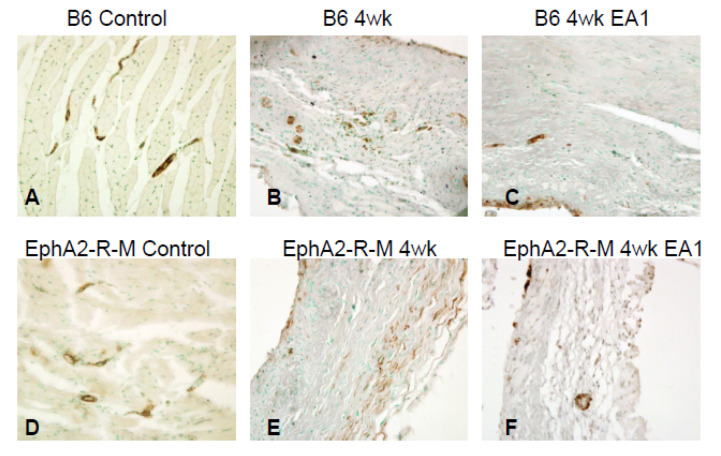
Active Myofibroblasts in Infarcted WT B6 (**A**–**C**) and EphA2-R-M (**D**–**F**) Mouse Hearts. Smooth muscle α-actin positive cells in uninjured B6 and EphA2 control heart tissue demonstrate expected staining of VSMCs in large vessels (200×). At 4 weeks post-MI, SMα-A positive myofibroblasts are visible in the infarcted myocardium in both B6 and EphA2, but to a much lesser extent in ephrinA1-Fc-treated B6 and EphA2-R-M mice (*n* = 3/group).

**Figure 7 ijms-21-05811-f007:**
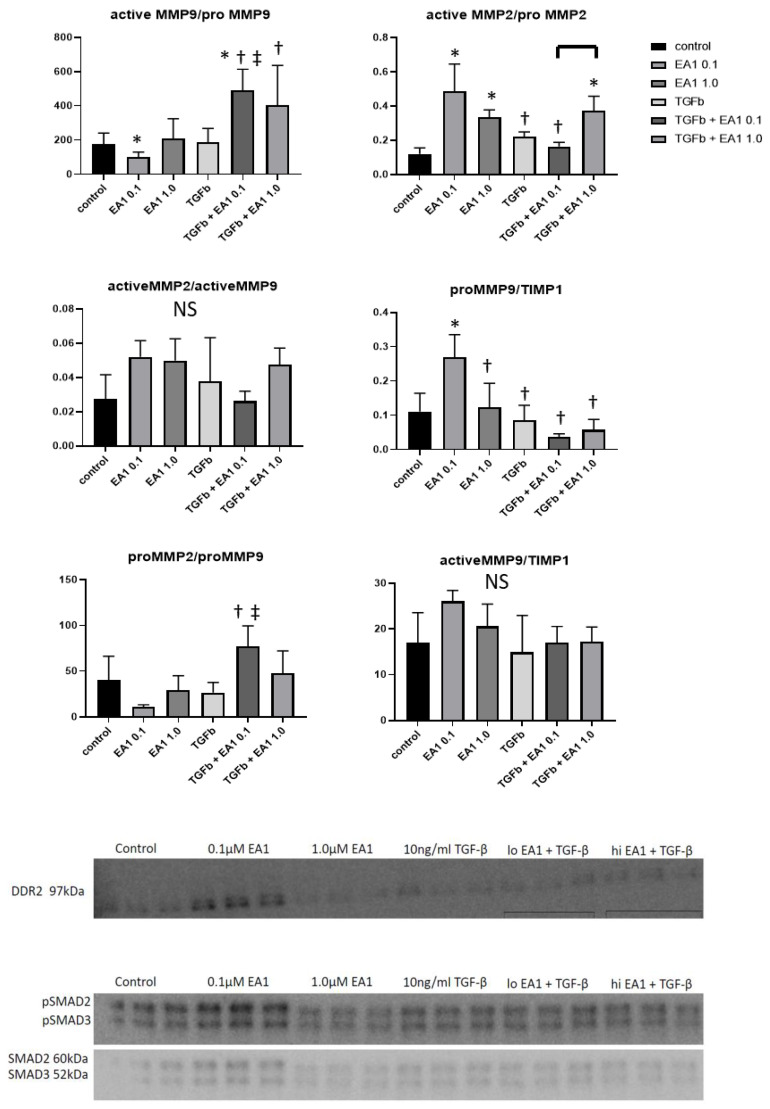
Expression Changes in MMP-2, MMP-9, TIMP1, DDR2 and (p)SMAD2/3/total SMAD2/3 in B6 and EphA2-R-M Mouse Hearts. MMP-9, MMP-2 and TIMP-1 expression in ephrinA1-Fc-treated +/-TGF-β primary cardiac fibroblasts are modulated in a dose-dependent manner (*n* = 3/group; significance: * vs. control, ^†^ vs. 0.1 µM EA1, ^‡^ vs. TGF-β). Western blots of DDR2 and (p)SMAD2/3/totalSMAD2/3 show dose-dependent changes in response to ephrinA1-Fc-treatment +/-TGF-β (NS).

**Table 1 ijms-21-05811-t001:** Morphometric measures in WT and EphA2-R-M mouse hearts.

Morphometric Parameters	WT B6 Uninjured (*n* = 10)	EphA2-R Uninjured (*n* = 10)	WT B6 4wk (*n* = 5)	EphA2-R 4wk (*n* = 12)	WT B6 4wk + EA1 (*n* = 8)	EphA2-R 4wk + EA1 (*n* = 5)
LVID_d_ (mm)	2.50 ± 0.10	2.82 ± 0.10	4.08 ± 0.30 *	3.58 ± 0.32 *	2.86 ± 0.08	3.64 ± 0.50
LVID_s_ (mm)	0.78 ± 0.06	1.51± 0.08	2.65 ± 0.27 *	2.65 ± 0.29 *	1.63 ± 0.08	2.66 ± 0.51
AWT (mm)	2.07 ± 0.09	2.07 ± 0.16	1.3 ± 0.12	2.34 ± 0.21	1.79 ± 0.11	1.71 ± 0.41
SWT (mm)	2.21 ± 0.19	2.02 ± 0.08	1.86 ± 0.23	2.18 ± 0.14	2.48 ± 0.13	2.74 ± 0.09 ^‡^
LVV_d_ (µL)	23.10 ± 2.43	75.92 ± 13.31	75.92 ± 13.32	60.81 ± 14.1	31.30 ± 2.28 ^†^	62.26 ± 21.04
LVV_s_ (µL)	1.21 ± 0.23	27.70 ± 7.18	27.70 ± 7.18	31.48 ± 9.24	7.77 ± 1.12 ^†^	32.59 ± 16.59
LV Area (mm^2^)	5.69 ± 0.9	6.07 ± 0.48	12.30 ±1.32 *	7.85 ± 1.35	11.66 ± 2.22	12.44 ± 2.82 ^‡^

*p* < 0.05 * vs. control, ^†^ vs. EphA2-R-M 4wk, ^‡^ vs. WT B6 4wk + EA1.

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
