# Peer review of "EphrinA1-Fc Attenuates Ventricular Remodeling and Dysfunction in Chronically Nonreperfused WT but not EphA2-R-M mice"

_ijms, 2020, doi:10.3390/ijms21165811_

Round 1

Reviewer 1 Report

A very good paper on the cardioprotective effect of Ephrin A1-Fc.

The Introduction is very informative and to the point. The methods description is clear and concise.

I have only a few suggestions :

The paragraph between lines 97-113 is hard to follow; the data might benefit from a table or an appendix for clarity.

The image quality is poor for the writing in panel G in figures 2, 3, and 4. The same for panel A in Figure 5.

C is missing from Figure 3. Also, an indication of what panels A-G represent should be included in the caption; in Figure 2 panel G is not included in the caption. The caption of Figure 4 should be used as an example. Same for Figures 5 and 6.

Given the fact that the discussion is quite long, the paper might benefit from a Conclusion subsection.

Also, I have two questions:

Why less survival with EphA2 treatment?

Why such a difference in EF at baseline between WT and EphA2-R-M?

Author Response

A very good paper on the cardioprotective effect of Ephrin A1-Fc.

The Introduction is very informative and to the point. The methods description is clear and concise.

Response: We thank the reviewer for the time and effort invested in evaluating our work and we are very enthusiastic that it was found to be well organized and received.

I have only a few suggestions :

The paragraph between lines 97-113 is hard to follow; the data might benefit from a table or an appendix for clarity.

Response: We have removed this paragraph and replaced it with a table as suggested.

The image quality is poor for the writing in panel G in figures 2, 3, and 4. The same for panel A in Figure 5.

C is missing from Figure 3. Also, an indication of what panels A-G represent should be included in the caption; in Figure 2 panel G is not included in the caption. The caption of Figure 4 should be used as an example. Same for Figures 5 and 6.

Response: We have adjusted/replaced these figures and added the information requested into the legends and hope that this clarifies any confusion or concerns.

Given the fact that the discussion is quite long, the paper might benefit from a Conclusion subsection.

Response: We heavily considered this suggestion but were unable to decide what headings to use that would be appropriate to assist the reader so we welcome suggestions.

Also, I have two questions:

Why less survival with EphA2 treatment?

Why such a difference in EF at baseline between WT and EphA2-R-M?

Response: These are excellent questions and although we do not have answers, we hypothesize that there may be signaling in other cells types (purkinje?) that may cause electrical disturbances and since this is a global knockout from birth, there may be developmental differences (cardiac ultrastructure, metabolism/mitochondrial function?) that may manifest in this experimental situation.

Reviewer 2 Report

Dear authors, thank you for the good research.

Please revise some minor corrections.

  1. Please make the table with the Morphometry (lines 96-113) for the better understanding.
  2. Please insert the scale bar and indicate the staining used as well as magnification in all the description of histological pictures (Fig 2AD, Fig 3AD, Fig 4AD, Fig 6AD)
  3. Figure 5A. Please make the Figure with higher resolution. It is not possible to understand the legend.
  4. Please indicate the number of animals in each group in 4.1 (Animals)
  5. Please insert the Ref to the published abstract with the preliminary (as far as I understand) results [Abstract 590: EphrinA1-Fc Attenuates Dysfunction and Fibrosis in Nonreperfused Myocardium at 4 weeks Post-MI in WT B6 but Not EphA2-R Mutant Mice]

Author Response

Dear authors, thank you for the good research.

Response: We are very pleased that the reviewer found merit in our work. Thank you for your time and effort in evaluating this manuscript and we hope that the revisions have improved its quality.

Please revise some minor corrections.

  1. Please make the table with the Morphometry (lines 96-113) for the better understanding. Response: We appreciate this suggestion and have replaced the paragraph with Table 1.
  2. Please insert the scale bar and indicate the staining used as well as magnification in all the description of histological pictures (Fig 2AD, Fig 3AD, Fig 4AD, Fig 6AD) Response: These modifications have been made.
  3. Figure 5A. Please make the Figure with higher resolution. It is not possible to understand the legend.Response: We apologize for this oversight and have replaced this figure. 
  4. Please indicate the number of animals in each group in 4.1 (Animals). Response: The number of animals used for each assay has been included in the legends and table/figure text.
  5. Please insert the Ref to the published abstract with the preliminary (as far as I understand) results [Abstract 590: EphrinA1-Fc Attenuates Dysfunction and Fibrosis in Nonreperfused Myocardium at 4 weeks Post-MI in WT B6 but Not EphA2-R Mutant Mice] Response: We thank the reviewer for catching this and have included this reference.